# Effect of Annealing on the Microstructure and SERS Performance of Mo-48.2% Ag Films

**DOI:** 10.3390/ma13184205

**Published:** 2020-09-22

**Authors:** Haoliang Sun, Xinxin Lian, Yuanjiang Lv, Yuanhao Liu, Chao Xu, Jiwei Dai, Yilin Wu, Guangxin Wang

**Affiliations:** 1School of Materials Science and Engineering, Henan University of Science and Technology, Luoyang 471023, China; lianxinxincn@163.com (X.L.); lyj_solovely@163.com (Y.L.); yh088232@163.com (Y.L.); chaoxu811@163.com (C.X.); dai978335921@163.com (J.D.); wyl1593571232@163.com (Y.W.); wgx58@126.com (G.W.); 2Collaborative Innovation Center of Nonferrous Metals Henan Province, Luoyang 471003, China

**Keywords:** Mo-48.2% Ag film, anneal, polyhedral Ag particles, SERS

## Abstract

Mo-48.2% Ag films were fabricated by direct current (DC) magnetron sputtering and annealed in an argon atmosphere. The effects of annealing on the surface morphology, resistivity and surface-enhanced Raman scattering (SERS) performance of Mo-48.2% Ag films were investigated. Results show a mass of polyhedral Ag particles grown on the annealed Mo-48.2% Ag films’ surface, which are different from that of as-deposited Mo-Ag film. Moreover, the thickness and the resistivity of Mo-48.2% Ag films gradually decrease as the annealing temperature increases. Furthermore, finite-difference time-domain (FDTD) simulations proved that the re-deposition Ag layer increases the “hot spots” between adjacent Ag nanoparticles, thereby greatly enhancing the local electromagnetic (EM) field. The Ag layer/annealed Mo-48.2% Ag films can identify crystal violet (CV) with concentration lower than 5 × 10^−10^ M (1 mol/L = 1 M), which indicated that this novel type of particles/films can be applied as ultrasensitive SERS substrates.

## 1. Introduction

Surface-enhanced Raman scattering (SERS) is a super-sensitive detection technique [1,2], which has become a powerful spectral method for identification and quantitative analysis of compounds [3]. However, the preparation of high-performance SERS substrate becomes crucial to the practical application of SERS technology due to the weak Raman signal [4,5]. The performance of SERS depends largely on the surface properties and materials of SERS substrates [6]. SERS substrates are usually made of noble metal (Ag, Au, Cu) nanomaterials, which can excite local surface plasmon resonance (LSPR) to generate strong extinction and scattering spectra [7]. In particular, SERS substrates composed of Ag nanostructures exhibit better SERS performance [8].

Currently, various Ag nanostructures, such as nanospheres [9], cubic [10], flower-like [11], triangles [12], nanowires [13], nanoneedles [14], nanorods [15], nanoshells [16], nanoclusters [17] and other polyhedral shapes [18], have been made by “top-down” or “bottom-up” assembly methods [19]. “Top-down” lithography with high-reproducibility and large-scale manufacturing has been widespread applied in industry [20]. However, the complicated synthesis or fabrication procedures cause inconvenience in common analysis [21]. Therefore, the relatively “bottom-up” method has developed greatly in recent years, for example, the solvent evaporation method [22], the electrodeposition method [23], Langmuir–Blodgett technology [24], the self-assembly method [25] and other methods [26]. The self-assembly technology, in particular, makes use of the surface functional groups, potential, morphology and other characteristics of nanostructures to spontaneously form dense and controllable nanostructures [27]. So far, it is still a focus in the research field of SERS substrate structures.

It is imperative to manufacture highly efficient SERS-active substrates in order to detect and identify trace harmful substances accurately and efficiently. Based on previous studies on the as-deposited Mo-Ag films and the annealed Ag-Zr films [28,29,30], plenty of self-formed Ag particles were found on the films’ surface, which leads to a strong Raman response. As is well known, annealing has an important influence on the microstructure, residual stress of the film. Therefore, we planned to adjust the particle size, morphology and quantity of self-formed Ag particles on the Mo-48.2% Ag films by annealing to obtain a more sensitive SERS performance.

## 2. Materials and Methods

Mo-48.2% Ag films were prepared by a direct current (DC) magnetron sputtering system (JCP-350). The composite targets consisted of a Ø 50 mm × 4 mm Mo target (99.99%) and four Ag sheets (5 mm × 5 mm × 1 mm) sputtered in high-purity argon (99.9999%).

Flexible polyimide (PI) substrates were cleaned with acetone, deionized water and ethanol for 15 min [28]. The pressure of the deposition chamber was pumped to 5 × 10^−4^ Pa, and pure argon was introduced to achieve deposition pressure (0.5 Pa). Sputtering power was controlled at about 100 W and pre-sputtered for 3 min [29]. Annealing was carried out at 160~360 °C, kept for 60 min in an argon atmosphere, and cooled in the furnace. Additionally, the annealed Mo-48.2% Ag films’ surface was covered with an Ag layer under the same experimental parameters as the Mo-48.2% Ag film.

The morphology, composition and phase of Mo-48.2% Ag films were characterized through field-emission scanning electron microscope (FE-SEM, JSM 7800, JEOL Ltd., Tokyo, Japan) with energy-dispersive spectroscopy (EDS) and X-ray diffraction (XRD, Bruker-AXS D8 Advance, SHIMADZU LIMITED, Kyoto, Japan). The resistivity of the Mo-48.2% Ag film was characterized by four-point probe resistance (RTS-8) and the 2D electric field distribution was simulated by the finite-difference time-domain (FDTD) method. The SERS signals of crystal violet (CV, Sinopharm Chemical Reagent Co., Ltd., Beijing, China) on the Mo-48.2% Ag films were characterized by Raman spectrometer (Invia Raman Microscope, Renishaw, Cambridge, UK).

## 3. Results and Discussion

### 3.1. X-ray Diffraction (XRD) Patterns of the Mo-Ag Films with Different Annealing Temperature

The XRD patterns of the Mo-Ag films before and after annealing are shown in Figure 1. The as-deposited Mo-Ag films exhibit weak diffraction peaks in the XRD pattern, inferring fine grains or amorphous structures in the Mo-Ag films at room temperature [30]. Compared with the as-deposited Mo-Ag films, the intensity of the Ag (111) diffraction peaks in the Mo-Ag films annealed at 260 °C and 360 °C increased obviously. The parameters of crystallites in the films annealed at different temperatures is calculated by the Bragg equation (2dsinθ=kλ) and Scherrer formula $\left(D=\frac{0.89\lambda }{\beta \cos\theta }\right)$. The calculated results of the Ag (111) diffraction peaks are listed in Table 1, indicating that annealing can promote the slow growth of Ag grains. Conversely, the intensity of the Mo (110) diffraction peaks in the Mo-Ag films decreases as the annealing temperature increases, which demonstrated that the Ag grains have an inhibitory effect on the growth of Mo grains [28]. Additionally, there are no diffraction peaks of Mo and Ag compounds in XRD patterns due to the extremely low solubility of Mo and Ag [31].

### 3.2. Morphology Characterization of Annealed Mo-Ag Films

Figure 2 shows the scanning electron microscope (SEM) images of the Mo-Ag films before and after annealing. It can be seen from Figure 2a that a large number of irregular nanoparticles grew on the as-deposited Mo-Ag films’ surface. Based on the previous transmission electron microscopy (TEM) and EDS analysis of the self-grown particles on the Mo-Ag films’ surface, it can be determined that these particles are Ag particles [28,30]. The structure of Ag particles/Mo-Ag films/PI is similar to that of Ag clusters/Si. However, the preparation condition and formation mechanism of Ag particles are completely different from Ag nanoclusters [17]. According to EDS analysis, the composition of Ag and Mo in the Mo-Ag film are 48.2% and 51.8%, respectively. The morphology and size of Ag particles on the annealed Mo-Ag film is obviously different from those on the as-deposited Mo-Ag film, as shown in Figure 2a–d. Compared with the Ag particles on the as-deposited Mo-Ag film, the size of Ag particles on the Mo-Ag film annealed at 160 °C is increased by 28 nm and the morphology becomes more regular. As annealing temperature rises to 260 °C and 360 °C, Ag particles have grown into polyhedrons with sharp edges and corners, as shown in Figure 2c,d. Meanwhile, the size and number of polyhedral particles have greatly increased because Ag atoms diffuse more actively. In addition, the composition of the Mo-Ag film before and after annealing is basically unchanged according to the EDS analysis in Figure 2d. Figure 2e_1_ is the element mapping of the Mo-48.2% Ag film annealed at 360 °C in Figure 2e. Figure 2e_2_,e_3_ show that element mapping of Mo and Ag distributed uniformly in the Mo-48.2% Ag film, respectively. Moreover, Figure 2e_3_ shows that the particles on the annealed 360 °C Mo-48.2% Ag film are Ag particles, which is consistent with the previous studies of as-deposited Mo-Ag film [28]. Because of the mutual inhibition of Mo and Ag grains in the Mo-Ag films [30,32], the distortion energy and residual stress are released through sufficient mass transport during the annealing process to form plenty of polyhedral particles on the Mo-48.2% Ag films.

### 3.3. Preparation and Raman Measurement of the Ag Layer/Annealed Mo-48.2% Ag Film

Figure 3 is a schematic diagram of the formation of Ag particles on the Mo-48.2% Ag film. The Mo-48.2% Ag films were prepared on PI substrates at room temperature, as shown in Figure 3I. According to previous studies [33,34], the internal stress of the film can be released by forming particles on the film. Therefore, numerous “worm-like” particles were formed on the as-deposited Mo-48.2% Ag films. Note that the morphology and size of the Ag particles on the annealed Mo-48.2% Ag films are significantly different from those of the as-deposited Mo-48.2% Ag films because of the stronger diffusion ability of Ag atoms at higher temperature, as shown in Figure 3II. Moreover, as the annealing temperature rises, the morphology of Ag particles changed from irregular particles on the as-deposited Mo-48.2% Ag film to polyhedral particles on the annealed Mo-48.2% Ag film. Simultaneously, the further relaxation of stress in the Mo-48.2% Ag film will grow some new small Ag particles. The formation mechanism of the polyhedral Ag particles on the annealed Mo-Ag alloy film is similar to that of the annealed Ag-Zr alloy film [29], but is different from that of hillocks observed in the Pt/Ti film [35]. An Ag layer can be deposited on the annealed Mo-48.2% Ag films in order to obtain large specific surface area, as shown in Figure 3III. Finally, the CV molecules adsorbed on the Ag layer/annealed Mo-48.2% Ag films are detected by a Raman spectrometer with 532 nm laser wavelength, as shown in Figure 3IV. The laser excitation energy, diffraction grid and spot are 5 mW, 1200 gr/mm and 2 μm, respectively. The scanning range and acquisition time are, respectively, 200–2000 cm^−1^ and 1 s.

### 3.4. Electrical Performance of the Annealed Mo-48.2% Ag Film

Figure 4a–d shows cross-sectional morphology of the Mo-Ag films before and after annealing. Obviously, the Ag particles are distributed on the Mo-48.2% Ag film, and the position relationship between Ag particles and the films is different from the hillock [36]. The thickness of the Mo-48.2% Ag film can be clearly seen from the cross-sectional morphology, and the film thickness gradually decreases as the annealing temperature increases as shown in Figure 4e. The reason is that the formation and growth of polyhedral Ag particles on the annealed Mo-Ag film consumes abundant Ag atoms in the vicinity of the film surface. The square resistance of the Mo-48.2% Ag film was characterized by the four-point probe resistance, and the results are shown in Figure 4f. Meanwhile, the resistivity of the Mo-48.2% Ag films gradually decreases as the annealing temperature rises, which can be ascribed to grain growth and reduction of the defects in the annealed films [37]. Moreover, the decrease of electron scattering in the film promotes the transmission of electrons, which leads to the decrease of resistivity and square resistance of the film.

### 3.5. Surface-Enhanced Raman Scattering (SERS) Activity of the Ag Layer/Annealed Mo-48.2% Ag Films

Figure 5a is the SEM image of the Mo-48.2% Ag film annealed at 360 °C after further sputtering the Ag layer. Deposition of an Ag layer on the annealed Mo-48.2% Ag film can not only significantly increase the Ag particle size and the surface roughness, but also decrease the gaps between Ag particles. Figure 5b is the FDTD simulation of the EM field intensity in the red circle region of Figure 5a. Obviously, compared with the annealed Mo-48.2% Ag film, the electric field intensity of the Ag layer/annealed Mo-48.2% Ag film is significantly enhanced. This phenomenon confirms that covering an Ag layer on the annealed Mo-48.2% Ag films can increase the number of “hot spots” and cause a near-field enhancement. In order to make the CV fully adsorbed on the SERS substrates, the Ag layer/annealed Mo-48.2% Ag films were soaked in a 10 mL CV solution for 1 h and dried naturally in the air before Raman testing. Figure 5c shows the Raman spectra of CV molecules adsorbed on the Ag layer/Mo-48.2% Ag films with different annealing temperatures, and the fingerprint bands in the spectra of CV molecules are observed [38,39]. Moreover, it is noted from Figure 5c that the Ag layer/Mo-48.2% Ag film annealed at 360 °C has the excellent SERS performance. Therefore, to further determine the SERS sensitivity of the Ag layer/Mo-48.2% Ag film annealed at 360 °C as SERS substrates, the SERS substrates were soaked in 5 × 10^−6^, 5 × 10^−7^, 5 × 10^−8^, 5 × 10^−9^ and 5 × 10^−10^ M CV solutions, respectively. Note that SERS spectra in Figure 5d show that Raman spectrum of CV solution with concentration of 10^−10^ M could still be clearly discerned successfully. In addition, the enhancement factor (EF) can be accurately calculate by following formula [40]:(1)$$\mathrm{EF}=\frac{I_{SERS}\;/\;C_{SERS}}{I_{Raman\;}/\;C_{Raman}}=\frac{I_{SERS}}{I_{Raman}}\times \frac{C_{Raman}}{C_{SERS}}=\frac{2376.216}{46.389}\times \frac{10^{-3}}{10^{-10}}=5.12\times 10^{8}$$
where I_SERS_ and I_Raman_ are the integrated intensities of the SERS signal of the Ag layer/Mo-48.2% Ag film (1622 cm^−1^) and pure PI substrate, respectively. C_SERS_ and C_Raman_ are the concentrations of CV adsorbed on the Ag layer/Mo-48.2% Ag film and PI substrate, respectively. The Ag layer/Mo-48.2% Ag films annealed at 360 °C as SERS substrates exhibit preeminent SERS activity to CV molecules, and the enhancement factor can reach $\;5.12\times 10^{8}$. In order to verify the repeatability of the Ag layer/Mo-48.2% Ag films as SERS substrates, we randomly selected 20 different points on the same sample for Raman testing, and the results are shown in Figure 5e. Additionally, the strongest characteristic peaks at 1622 cm^−1^ in the CV molecular spectrum were selected to evaluate the repeatability of the SERS substrate, as shown in Figure 5f. Obviously, the intensity of the Ag layer/Mo-48.2% Ag film at 20 points of the 1622 cm^−1^ peaks with 5 × 10^−8^ M CV solutions is very close to 8000, which indicates that the film has good uniformity and repeatability.

## 4. Conclusions

Ag layer/annealed Mo-48.2% Ag films were fabricated on PI substrate by DC magnetron sputtering and atmosphere annealing. The Ag particles on the as-deposited Mo-48.2% Ag films are irregular, but the Ag particles on the annealed Mo-48.2% Ag film are gradually transformed into polyhedrons with sharp edges and corners. Moreover, the film thickness and resistivity gradually decrease as the annealing temperature increases. Furthermore, covering an Ag layer on the annealed Mo-48.2% Ag film can not only significantly increase the Ag particle size and the surface roughness, but also decrease the gaps between Ag particles, resulting in a significant enhancement of the local EM field. The Ag layer/Mo-48.2% Ag films annealed at 360 °C as SERS substrates show high sensitivity of the enhancement factor (EF) up to $5.12\times 10^{8}$ with the detection limit concentration of CV lower than 10^−10^ M. As a high-sensitivity fingerprint identification tool, the Ag layer/Mo-48.2% Ag films as SERS substrates have shown great application potential in biomedicine, environmental monitoring, food and drug safety and other fields.

## Figures and Tables

**Figure 1 materials-13-04205-f001:**
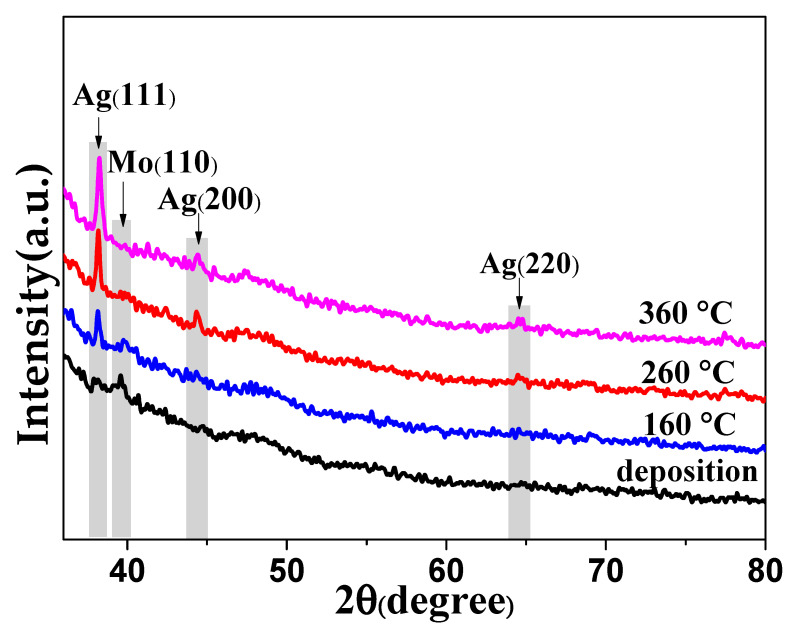
X-ray diffraction (XRD) patterns of the Mo-Ag films before and after annealing.

**Figure 2 materials-13-04205-f002:**
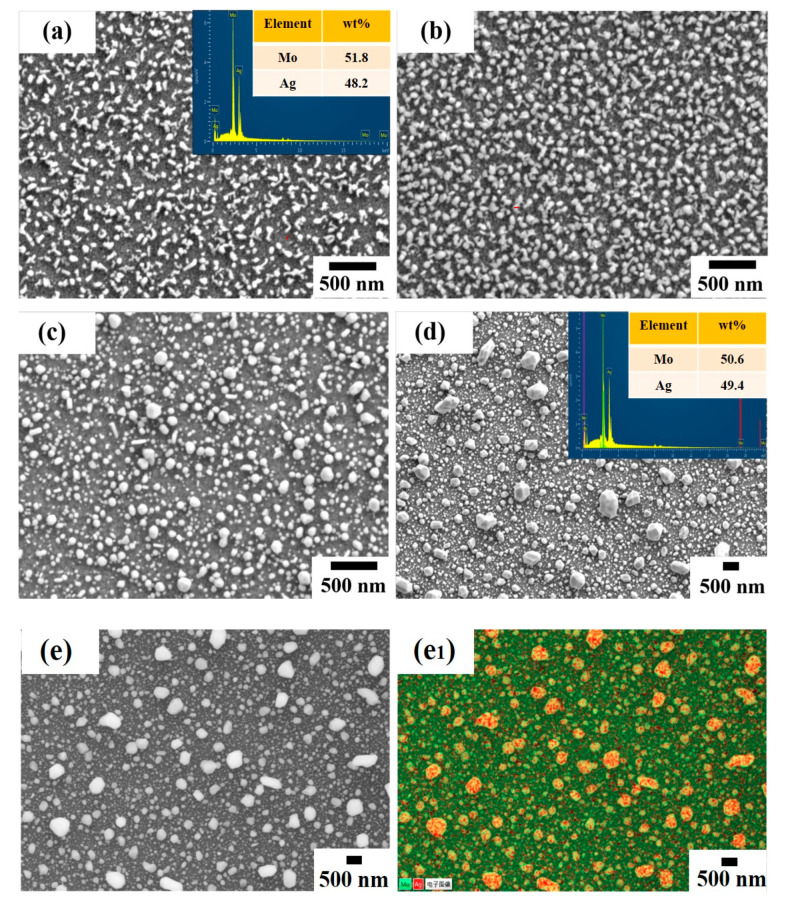
The scanning electron microscope (SEM) images of the Mo-48.2% Ag films before and after annealing: (**a**) room temperature; (**b**) 160 °C; (**c**) 260 °C; (**d**) 360 °C; (**e**) the electronic image of Figure 2d; (**e_1_**) element mapping of Figure 2e; (**e_2_**) Mo mapping of Figure 2e_1_; (**e_3_**) Ag mapping of Figure 2e_1_.

**Figure 3 materials-13-04205-f003:**
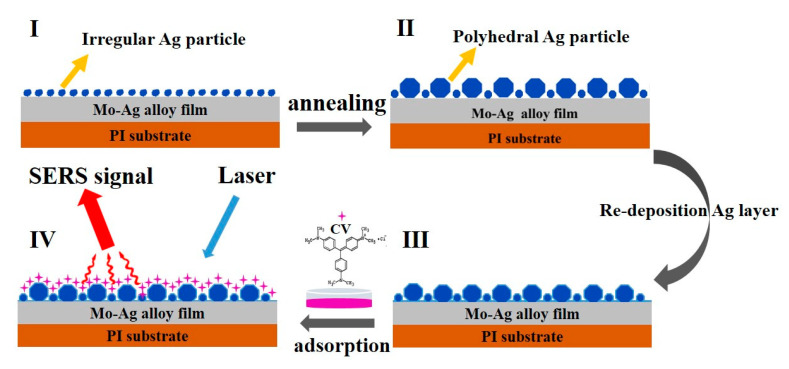
Schematic diagram of the Ag layer/annealed Mo-48.2% Ag film formation: (**I**) Mo-48.2% Ag films were prepared on PI substrate at room temperature; (**II**) the Mo-48.2% Ag film annealed in argon atmosphere; (**III**) an Ag layer was further sputtered on the annealed Mo-48.2% Ag films; (**IV**) the surface-enhanced Raman scattering (SERS) performance of the Ag layer/annealed Mo-48.2% Ag film was measured.

**Figure 4 materials-13-04205-f004:**
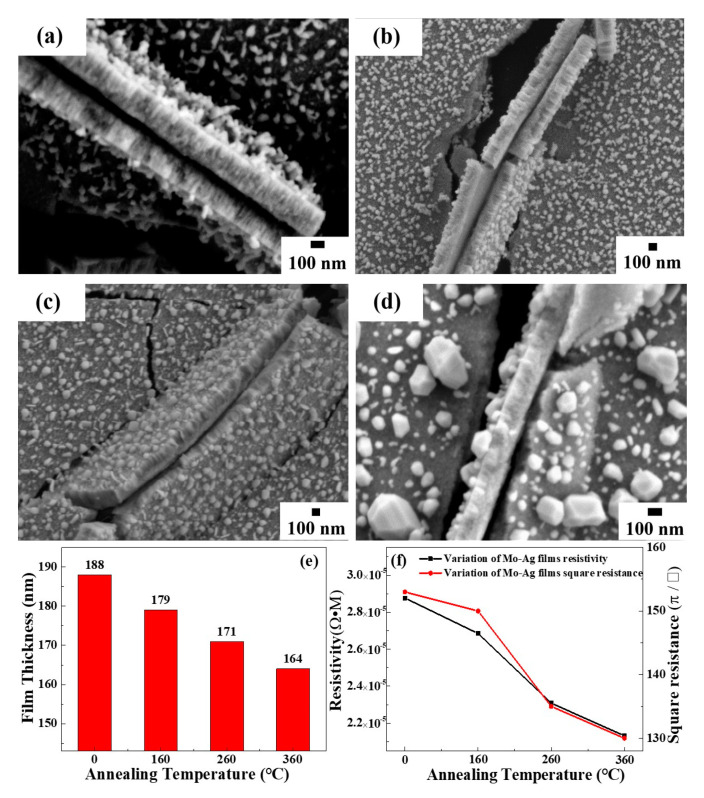
Cross-sectional morphology of Mo-48.2% Ag films annealed before and after annealing: (**a**) room temperature; (**b**) 160 °C; (**c**) 260 °C; (**d**) 360 °C; (**e**) variation of the film thickness with annealing temperature; (**f**) variation of square resistance and resistivity of Mo-48.2% Ag film with annealing temperature.

**Figure 5 materials-13-04205-f005:**
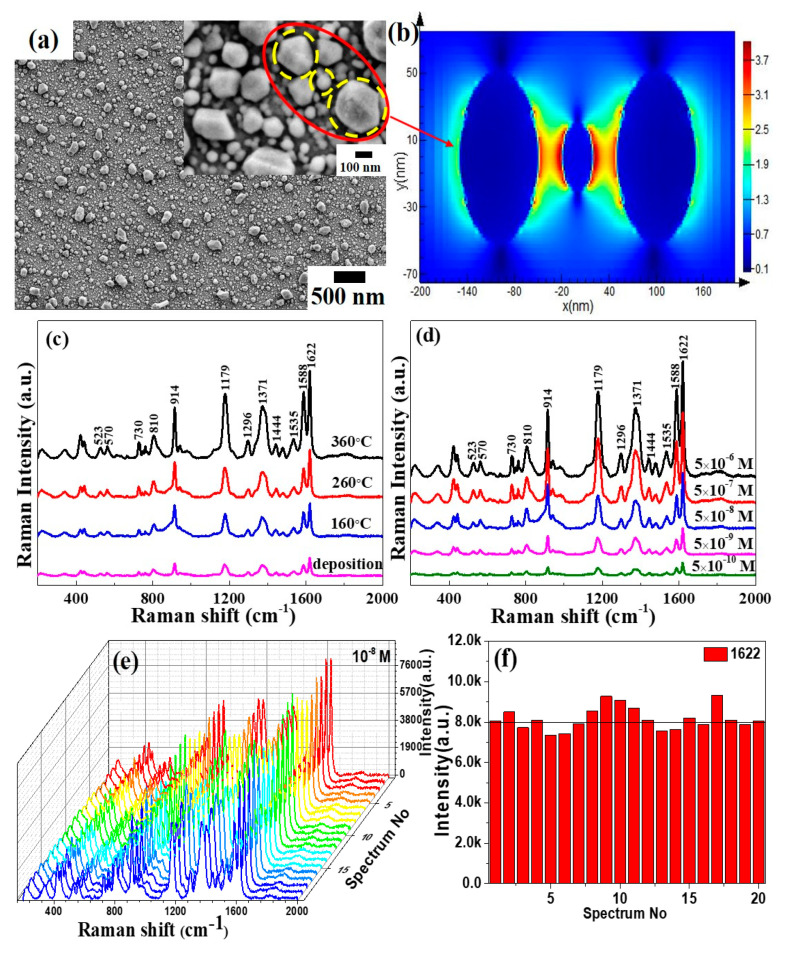
(**a**) SEM image of the Ag layer/Mo-48.2% Ag films annealed at 360 °C; (**b**) finite-difference time-domain (FDTD) simulations of the EM field intensity in the red circle region of Figure (**a**); (**c**) SERS spectra of the Ag layer/Mo-48.2% Ag films with different annealing temperatures; (**d**) SERS spectra of the Ag layer/Mo-48.2% Ag films annealed at 360 °C in different concentrations of CV solution; (**e**) SERS spectra of 20 randomly selected locations in the Ag layer/Mo-48.2% Ag films annealed at 360 °C; (**f**) variation of the Raman spectra intensity at 1622 cm^−1^ peaks in (**e**).

**Table 1 materials-13-04205-t001:** The parameters of crystallites in the films annealed at different temperatures.

Annealing Temperature	2θ (Degree)	d (Å)	β (Degree)	D (nm)
Room temperature	39.573	2.275	0.319	26
160	38.250	2.351	0.268	31
260	38.148	2.357	0.258	32
360	38.271	2.350	0.249	33

The X-ray wavelength λ is 0.15406 nm and the Bragg diffraction angle is θ. Interplanar spacing, full width at half maximum and mean dimension of crystallites are represented by d, β, and D, respectively.

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
