# Peer review of "Effect of Annealing on the Microstructure and SERS Performance of Mo-48.2% Ag Films"

_materials, 2020, doi:10.3390/ma13184205_

Round 1
Reviewer 1 Report
Manuscript number: Materials-923394
Title: Effect of annealing on the microstructure and SERS performance of Mo-48.2 % Ag films
The authors described the effects of annealing on the surface morphology, resistivity and SERS performance of the Mo-48.2 % Ag films. The research is a continuation of the published article entitled “Room temperature self-assembled Ag nanoparticles/Mo-37.5% Ag film as efficient flexible SERS substrate”. So, the novelty and originality of the study are not high.
I have the following observations:
- A reference for deposition method is required.
- The mean dimension of crystallites can be calculated from XRD patterns.
- “3.3 Formation mechanism” - it is not really a “mechanism”, in the chemical sense of a particle formation.
Reviewer 2 Report
The work presented here involves the deposition of a Mo Ar thin films that are then annealed at different temperatures and characterised using SEM, EDX, XRD, 4 point probe and SERS. The authors have found that annealing caused a reduction in resistance, improvement in Raman response and increase in nanoparticle size while reducing film thickness.
The work is reasonably interesting and those in the field will find the results useful. However, I have some concerns that need to be addressed before it is published.
My main concern is that the repeatability of the work is not commented on, discussed, or shown in regard to different areas of the sample and across multiple repeat samples. It is possible the trends shown in the results are just features of non-uniform film deposition. The authors should have made multiple samples and carried out the annealing and characterisation on all of them to check for consistency. This will also allow of error bars to added to the graphs and allow for discussion of reproducibility of the method.
In the SEM images should include a zoomed out wide image that shows the uniformity of the film over the whole substrate.
The XDR is very noise and it is difficult to see the changes in features, especially the Ag 200 and Ag220 peaks. If more repeats are taken then they could be averaged to remove the noise.
The method section read more like a dissertation that a Journal Article. This should be addressed.
In the conclusion, there should be a discussion of potential practical applications for the findings.
Round 2
Reviewer 2 Report
Thank you to the authors for replying to my comments. I believe this work can now be published in this journal.
